# Estimating treatment costs for uncomplicated diabetes at a hospital serving refugees in Kenya

Lizah Masis[1¤a], Lucy Kanya[1]*, John Kiogora[2], Lilian Kiapi[3], Caitlin Tulloch[4], Ahmad Hecham Alani[3¤b]

**1** Department of Health Policy, London School of Economics and Political Science, London, United Kingdom, **2** International Rescue Committee, Nairobi, Kenya, **3** International Rescue Committee, London, United Kingdom, **4** International Rescue Committee, New York City, NY, United States of America

¤a Current address: Financing Alliance for Health, Nairobi, Kenya
¤b Current address: Scout Data, London, United Kingdom
* L.kanya@lse.ac.uk

**Data Availability Statement:** All relevant data are within the manuscript and Supporting Information files.

## Abstract

Diabetes mellitus (DM) is increasing markedly in low- and middle-income countries where over three-quarters of global deaths occur due to non-communicable diseases. Unfortunately, these conditions are considered costly and often deprioritized in humanitarian settings with competing goals. Using a mixed methods approach, this study aimed to quantify the cost of outpatient treatment for uncomplicated type-1 (T1DM) and type-2 (T2DM) diabetes at a secondary care facility serving refugees in Kenya. A retrospective cost analysis combining micro- and gross-costings from a provider perspective was employed. The main outcomes included unit costs per health service activity to cover the total cost of labor, capital, medications and consumables, and overheads. A care pathway was mapped out for uncomplicated diabetes patients to identify direct and indirect medical costs. Interviews were conducted to determine inputs required for diabetes care and estimate staff time allocation. A total of 360 patients, predominantly Somali refugees, were treated for T2DM (92%, n = 331) and T1DM (8%, n = 29) in 2017. Of the 3,140 outpatient consultations identified in 2017; 48% (n = 1,522) were for males and 52% (n = 1,618) for females. A total of 56,144 tests were run in the setting, of which 9,512 (16.94%) were Random Blood Sugar (RBS) tests, and 90 (0.16%) HbA1c tests. Mean costs were estimated as: $2.58 per outpatient consultation, $1.37 per RBS test and $14.84 per HbA1c test. The annual pharmacotherapy regimens cost $91.93 for T1DM and $20.34 for T2DM. Investment in holistic and sustainable non-communicable disease management should be at the forefront of humanitarian response. It is expected to be beneficial with immediate implications on the COVID-19 response while also reducing the burden of care over time. Despite study limitations, essential services for the management of uncomplicated diabetes in a humanitarian setting can be modest and affordable. Therefore, integrating diabetes care into primary health care should be a fundamental pillar of long-term policy response by stakeholders.

**Funding:** The authors received no specific funding for this work.

**Competing interests:** The authors have declared that no competing interests exist.

## Introduction

Diabetes mellitus is a leading contributor of non-communicable diseases (NCDs) deaths globally and most markedly in low- and middle-income countries (LMICs) [1, 2]. In 2020, approximately half a billion people had diabetes globally, with about 80% of these in LMICs [3]. The costs of managing diabetes are projected to rise to half a trillion US dollars (USD) world-wide by 2030 [4]. In addition to the economic pressures, LMICs continue to suffer the largest burden of extended humanitarian crises, necessitating the need for NCDs to be addressed [5]. These hardships have placed pressure on humanitarian organizations such as the International Rescue Committee (IRC) to extend their programs and treat such long-term conditions. However, the exact diabetes burden in humanitarian settings remains unknown [6]. The humanitarian context also poses unique challenges, including health systems disruption, traumatic injuries, and deterioration of living conditions, all of which worsen existing chronic conditions [7].

There is an absolute dearth of evidence on economic evaluations in such settings [8]. A systematic review published in 2015 on the effectiveness of NCDs interventions within humanitarian contexts identified eight studies conducted over the last 35 years [5]. Only three of these focused on diabetes and none were conducted in Africa. Further, none of the studies focussed on the cost of diabetes management in the humanitarian landscape—further highlighting the possibility that investments in NCDs may have been economically inefficient in light of the paucity of evidence [5, 9]. A later review in 2020 also stated that NCDs are still under-recognized in the humanitarian literature [10].

The Dadaab complex was established in 1991 and is one of the largest refugee camps in the world. It is located in North-East Kenya, and consists of 3 camps; Dagahaley, Ifo and Hagadera [11]. According to the IRC's Kenya Sheet, a total of 83,861 refugees mainly from Somalia lived in the complex in 2018. Healthcare services and community outreach programs are supervised by the United Nations High Commissioner for Refugees (UNHCR), and freely delivered at 16 health posts, four hospitals and one maternity centre [12]. According to UNHCR 2018 annual report, a total of 79,235 consultations were provided in the camp through all service delivery points; mainly for communicable diseases (58.6%) with the remaining (41.4%) distributed between NCDs, mental health and injuries. For NCDs consultations, the majority were for cardiovascular diseases (45.7%), followed by endocrine/metabolic disorders including diabetes (21.4%) [13]. The crude mortality rate in the camp is 0.1/1000/month (standard <0.75) [13]. Availability of services within the camp is restricted due to: safety concerns, abductions of aid workers, improvised explosive devices and attacks on refugee leaders and police [12]. The results of this study provide much-needed evidence on the investment in holistic and sustainable NCDs services within the humanitarian context for use by policymakers and stakeholders. Moreover, this study will contribute to the global knowledge on economic evaluations of NCDs interventions within humanitarian contexts.

## Materials and methods

### Study setting

The International Rescue Committee (IRC) Hagadera hospital is located within the Hagadera Refugee camp. The 140-bed secondary care facility also provides primary healthcare (PHC) services and supports four primary care health posts. NCD patients and complicated cases are referred from the primary care health posts to the hospital. The camp's health and nutrition services are provided primarily by the IRC. The facility was selected for this analysis due to the high caseload (more than 25,000 outpatient visits in 2017), and the provision of a sufficient

range of NCD services at the facility-IRC's Kenya factsheets for 2017 and 2018. It is estimated that the Hagadera refugee camp has nearly 1,000 patients with diabetes under follow-up in 2018. However, no previous diabetes prevalence studies have been reported in the camp.

## Study population

Utilization data was collected for adult patients who were managed for T1DM and T2DM in accordance with the World Health Organization (WHO) International Classification of Diseases (ICD-10) at the outpatient department (OPD) in 2017 [14]. Semi-structured short interviews (15–20 minutes) with a purposive sample of three administrative staff (hospital matron, health manager and a human resources officer), and seven health care professionals (HCPs) directly involved in diabetes management (two medical officers, one clinical officer, two laboratory technologists, and two pharmacists) were conducted between July-August 2018.

## Study design, methodology and framework

A retrospective cost-analysis was employed from a provider perspective as the IRC financed all health care services. The IRC's Systematic Cost Analysis (SCAN) tool was used to estimate costs per outputs. This tool enables the use of routinely gathered finance data to assess the cost and cost-efficiency of public health programs through estimating the financial cost per output [15]. The mixed-methods study design combined micro-costing and gross-costing approaches as per the trade-off methodology recommended by Hendriks et al [16]. Micro-costing is a method of cost estimation that allows for an accurate assessment of the economic costs of health interventions; by assessing the amount of each resource used bottom-up (e.g., consumables). Gross or top-down costing distributes a total budget to specific services such as OPD visits. Micro-costing was employed for elements that accounted for a substantial proportion of costs due to its high degree of detail [17, 18]. Gross-costing was applied for overheads as they require relatively fewer resources [16, 19]. A similar framework is endorsed by the Global Health Cost Consortium (GHCC) [18].

A care pathway was mapped out for patients with uncomplicated diabetes and relevant data on direct/indirect medical costs were collected. The study focused on diagnostic visits at the OPD including consultation, laboratory investigations, and pharmacotherapy. Labor, capital, medications/consumables, and overhead costs were the expense categories for healthcare activities (HCAs). Donated resources were valued at prevailing market prices to capture the true economic cost. For costing purposes, capital assets were assumed to have a useful life of more than a year, and a discount rate of 3% was adopted in line with literature recommendations [18, 20].

## Data collection

Direct/indirect medical costs were obtained from various sources details of which are presented in S1 Table. Broad cost categories are shown in (Fig 1). Data collection was carried out in four phases: (i) development of clinical pathway (ii) identification of related HCAs (iii) identification of cost items and (iv) cost items measurement, valuation, and allocation. The total number of patients utilizing each HCA was recorded to allocate services to diabetes care. Stakeholder interviews determined inputs required for diabetes care, and estimated staff time allocation.

## The clinical pathway for the care of uncomplicated diabetes

The pathway was developed based on comprehensive diabetes management guidelines endorsed by the government, WHO, and international organizations (Fig 2) [21–24]. HCPs

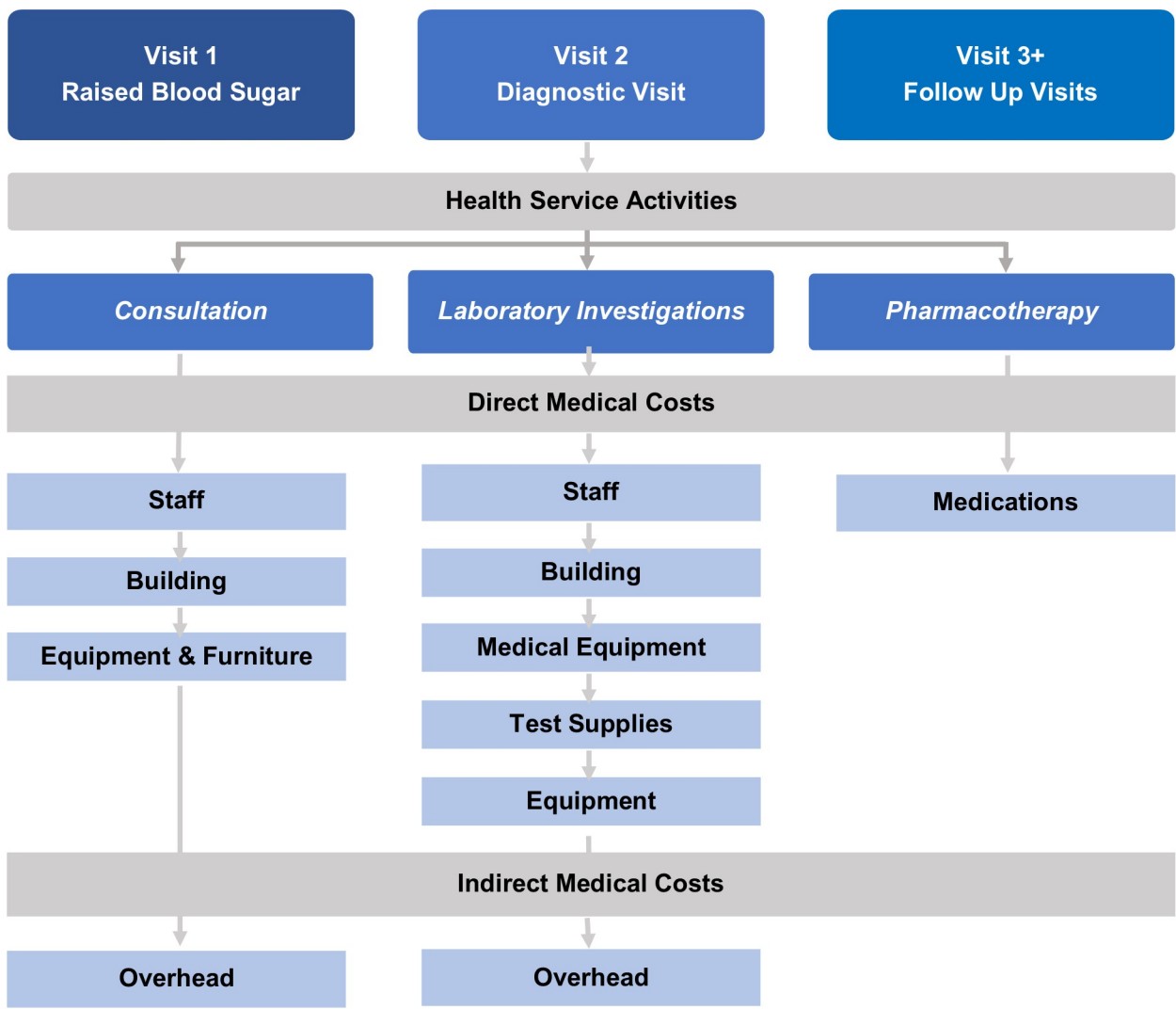

**Fig 1. Health service activities and direct/indirect cost items relevant for uncomplicated diabetes in the outpatient department.** Costs of follow-ups at the level of health posts or community were not considered.

interviews included questions to ensure guidelines representation at the hospital, in addition to the type and number of personnel and equipment used for diabetes care.

## Measurement, valuation and allocation of costs to the health service activities

**Staff costs.** Staff costs included in the analysis were for medical and clinical officers, laboratory technologists and attendants, nurses and medical assistants among others. These costs included salaries and fringe benefits (e.g., hardship allowances, pensions, medical insurance). As accommodation was provided freely, the equivalent rental value was used as a proxy. Based on interviews and activity log-sheets, real productive work hours and time spent in care were estimated and verified through time and motion studies [25, 26]. Administrative staff were interviewed for details on work shift-cycles and leave policies. Costs per annum and unit were proportional to the time used for each HCA.

**Fig 2. Clinical pathway for care of uncomplicated diabetes patients.**

**Capital costs.** A list of key furniture and equipment including replacement values and building repair expenses incurred in 2017 was obtained. Instead of rental cost, a building replacement value was applied for true costing. The number of tests performed on equipment per year for diabetes patients versus others (if shared equipment) was considered. As capital assets are exhausted by usage, their economic useful life was costed in accordance with the WHO-CHOICE program [27]. The annualized costs were calculated based on prevailing interest rate and useful life years shown in Table 1. Building costs were proportional to actual spaces utilized. Diabetes exclusive equipment were fully allocated to HCAs with appropriate proportions calculated for general furniture/equipment.

**Cost of drugs and consumables.** Unit prices included the costs of annual treatment regimens; (Insulin Mixtard 30/70 and syringes, 500mg Metformin and/or 5mg Glibenclamide), and stocks of laboratory tests/supplies of <2 years of useful life. Since Random/Fasting blood sugar (RBS/FBS) tests are conducted in a similar manner except for fasting status, both were referred to as "RBS". Notably, FBS testing is widely used for screening due to its high accuracy, compared to RBS testing [28]. The hemoglobin A1c (HbA1c) test is an indicator of blood sugar status over a 3-month period [28]. Since individual patient utilization data were unobtainable, overall service statistics were used to annualize costs.

**Table 1. Annualization factors[a].**

| Useful Life (Years) | Discount Rate (3%) | Useful Life (Years) | Discount Rate (3%) |
|:---:|:---:|:---:|:---:|
| 1 | 0.971 | 16 | 12.561 |
| 2 | 1.913 | 17 | 13.166 |
| 3 | 2.829 | 18 | 13.754 |
| 4 | 3.717 | 19 | 14.324 |
| 5 | 4.58 | 20 | 14.877 |
| 6 | 5.417 | 21 | 15.415 |
| 7 | 6.23 | 22 | 15.937 |
| 8 | 7.02 | 23 | 16.444 |
| 9 | 7.786 | 24 | 16.936 |
| 10 | 8.53 | 25 | 17.413 |
| 11 | 9.253 | 26 | 17.877 |
| 12 | 9.954 | 27 | 18.327 |
| 13 | 10.635 | 28 | 18.764 |
| 14 | 11.296 | 29 | 19.188 |
| 15 | 11.938 | 30 | 19.6 |

[a] Adapted from UNAIDS. Manual for costing HIV facilities and services.

**Overhead costs.** Two broad categories of indirect medical costs were perceived: local support costs incurred, and the high-level support incurred by the IRC's Kenya Country Program and headquarters. For local support, administrative personnel, security, housekeeping, office supplies, utilities, communications, transportation, facilities maintenance, fuel, and training costs were all considered. Equipment/building costs incurred by the general departments were also incorporated. High-level support costs included elements such as administrative staff costs and travel expenses. Organizational expenditure data was obtained from the IRC's financial records.

## Data management and analysis

Annual and unit costs were calculated by summing-up the yearly costs of expense categories. The calculated outputs were reported for both diabetes types by OPD consultation, laboratory investigations, and both monthly/annual pharmacotherapy regimens. Costs incurred before 2017 were adjusted using the GDP deflator [29]. An average exchange rate of KES 103.37 = USD 1 was used for reporting in USD [30]. A discount rate of 9% was determined from the Central Bank of Kenya [31, 32]. Data was processed and analyzed via Excel, version 14.1.3 (Microsoft Corp). Deterministic sensitivity analyses were applied to investigate uncertainty around the following parameters: (i) mark-up percentage for both OPD consultations and laboratory investigations (ii) discount rate and (iii) staff costs [17, 33]. Other uncertainty considerations such as model and generalizability were not regarded [34]. Ethical approvals were obtained from the London School of Economics & Political Science Institutional Review Board (UK)—reference #000717, and Amref Health Africa Ethics & Scientific Review Committee (Kenya)—protocol #P480-2018. Written consents were obtained from all facility staff interviewed including HCPs who were present during the time and motion studies.

## Results

## Main results

The OPD was staffed by one medical officer, a clinical officer, a nurse, and a medical assistant. The total number of OPD consultations in 2017 was 13,177 for males, and 13,517 for females

**Table 2. Patient characteristics and utilization[a].**

| Characteristic | n (%) |
|---|---|
| Total population covered by the hospital and four health posts | 42,435 |
| Total population covered by hospital | 16,987 (100) |
| Females | 8,588 (50.56) |
| Males | 8,399 (49.44) |
| T2DM patients | 331 (91.94) |
| Females | 195 (58.91) |
| Males | 136 (41.09) |
| T1DM patients | 29 (8.06) |
| Females | 21 (72.41) |
| Males | 8 (27.59) |
| **Utilization** | **n (%)** |
| Consultations in OPD | 26,694 (100) |
| Diabetes consultations in OPD | 3,140 (11.76) |
| Laboratory tests | 56,144 (100) |
| RBS tests | 9,512 (16.94) |
| HbA$_{1c}$ tests | 90 (0.16) |

[a]All data are for full year 2017

(n = 26,694), with a monthly average of 2,225. Patient characteristics and utilization data are presented in Table 2. While a significant number of patients are seen at the health posts and/or within the catchment population, only hospital visits were recorded. According to the IRC NCD indicators report (2017), the total number of outpatient consultations across the hospital and health posts was 158,965.

Of the 26,694 OPD consultations, a total of 3,140 were for diabetes (new visits and revisits); 1,522 and 1,618 for males and females respectively, with an average of 262 consultations per month. Both RBS/FBS and HbA1c tests were available for laboratory investigations. Analysis of the included 3,140 consultations resulted in 360 unique patients' majority of whom had T2DM (91.94%, n = 331), and only 29 (8.06%) had T1DM. A total of 56,144 tests were run, of which 16.94% (n = 9512) were RBS tests, with only 0.16% (n = 90) HbA1c tests. An average cost of $2.58 per diabetes consultation, regardless of type, was estimated after considering the major cost drivers. Other unit costs are reported in Table 3. Considering the number of diabetes patients, the total pharmacotherapy cost per annum for both types was calculated as $9,306.58. Co-prescription and dispensing costs were not included. Visits involving HbA1c tests were more expensive compared to RBS tests. A breakdown of visits by diabetes type and test established that T1DM was more expensive to treat than T2DM. Visits that involved HbA1c tests were also more expensive compared to RBS tests visits. Results were robust to all base case assumptions tested in the one-way sensitivity analyses.

## Annual and average unit costs of health service activities

The distribution of estimated economic costs is shown in S2 Table. Major cost drivers included personnel, test consumables and overhead costs. The annual cost of diabetes consultations was $8,099.94. The unit cost of a HbA1c test was nine times higher than the RBS test at $14.84 (RBS: $1.59). Annual costs for the latter test were higher at $13,098.72 compared to $1,335.57 (HbA1c tests) due to the frequency of the tests. The annual costs of pharmacotherapy regimens were $91.93 and $20.34 for T1DM and T2DM, respectively ($7.66 versus $1.69 per month).

**Table 3. Estimated unit costs in USD.**

| Costs of Health Service Activities | | | |
|---|---|---|---|
| Average Cost of Diabetes Consultation | $2.58 | Annual Cost of Diabetes Consultations (n = 3,140) | $8,099.94 |
| Average Cost of RBS test | $1.37 | Annual RBS tests (n = 9,512) | $13,098.72 |
| Average Cost of HbA$_{1c}$ test | $14.84 | Annual HbA$_{1c}$ tests (n = 90) | $1,335.57 |
| Monthly Cost of T1DM Medications Regimen | $7.66 | Annual Cost of T1DM Medications Regimen | $91.93 |
| Monthly Cost of T2DM Medications Regimen | $1.69 | Annual Cost of T2DM Medications Regimen | $20.34 |
| **Costs of Pharmacotherapy** | | | |
| Annual Cost of T1DM drugs | | $2,574.04 | |
| Annual Cost of T2DM drugs | | 6,732.54 | |
| **Total Costs by Diabetes Type (Including OPD consultation, test and pharmacotherapy)** | | | |
| Cost of T1DM + RBS test | | $11.83 | |
| Cost of T1DM + HbA$_{1c}$ test | | $25.08 | |
| Cost of T2DM + RBS test | | $5.86 | |
| Cost of T2DM + HbA$_{1c}$ test | | $19.11 | |

For the 360-patients included in this analysis, the annual cost of all drugs amounted to $9,306.58 ($2,574.04 for T1DM and $6,732.54 for T2DM).

## Sensitivity analysis

Results of the sensitivity analyses were robust to base case assumptions. Treatment costs were not significantly impacted by changes in the various parameters. For example, when the mark-up percentage was assumed to be similar for all services, an alternative assumption that consultations consumed fewer overhead costs—since patients are seen in the OPD- resulted in a 2.53% decrease in the total cost of consultations and laboratory tests. When more overheads were assumed (such as procurement, transportation etc.), the mark-up percentage increased by 4.51%. Additionally, when the discount rate was reduced to 3% as recommended for LMICs, along with the local discount rate for international comparisons, the resulting decrease in costs was negligible at 1.89% [18, 20, 35]. Furthermore, the Kenyan government placed a 14% cap on interest rates before and throughout 2017, which when adopted in the analysis increased the results by 3.69% [31]. To allow for variation in some factors that may affect staff cost, such as travel allowance and accommodation costs, a sensitivity analysis varying staff costs by ±10% yielded changes only by 3.71%.

## Discussion

Published literature shows that in the coming decades, LMICs will bear the greatest burden of diabetes, doubling the pressure on already fragile health systems [36–38]. This is further compounded by serious concerns surrounding the harmful impact of COVID-19 on people living with NCDs who are more susceptible to severe illnesses [39]. With inadequate surveillance or access to NCDs management, the risks of extreme outcomes and mortalities are increased [39]. Humanitarian settings within LMICs face additional challenges related to treatment access. However, there has been limited empirical research on the magnitude of NCDs, including cost analyses or economic evaluations in such settings [5, 40]. This hinders evidence-informed decision making and potentially leads to cost and efficiency losses. This study estimated the cost of treating uncomplicated diabetes within an OPD setting at the IRC Hagadera hospital in Kenya. The facility primarily caters to the health needs of Somali refugees. To the best of our knowledge, this is the first study to estimate the specific costs of treating uncomplicated diabetes in a humanitarian setting [8].

Of the 360 diabetics treated at the facility in 2017, 8.06% had T1DM while 91.94% had T2DM. The estimated unit costs of HCAs vary but not substantially. T1DM patients are underrepresented in this study which likely impacted their relevant estimated costs. However, higher volume activities such as RBS tests per year (n = 9,512) cost less at $1.59 per test than lower volume activities such as the HbA1c test (n = 90) at $14.84 per test. The HbA1c test was the main cost driver for both diabetes types, followed by consultation costs at $2.58 per visit (similar across both types). At an estimated unit cost of $7.66 for a monthly regimen, T1DM drugs were the second highest cost driver when a RBS test was used. The estimated unit costs for the treatment of T2DM were remarkably lower when compared to T1DM. This analysis established that essential basic services can be modest and affordable for the management of uncomplicated diabetes. The clinical pathway used provided a clear step-by-step view of where PHC services are consumed, and costs incurred, which enables calculations to be adjusted to different scenarios (e.g., allowing for more/less testing frequency, depending on access/ population needs). Several studies have shown that the economic cost of diabetes is underestimated when complications are excluded [41–45]. For instance, a cost-of-illness study concerned with diabetes financing from patients' out-of-pocket (OOP) spending reported wide variations in the estimated costs of care across several African countries and between the public and private healthcare systems. Notably, the OOP treatment costs soared when diabetes complications are present [46]. However, the presence of such complications/co-morbidities and the patient's perspective were beyond the study's scope. One-way sensitivity analyses were carried out to address economic uncertainty and the results were robust to all base assumptions.

There is limited empirical evidence on the cost of diabetes management broadly and none in the humanitarian context of Sub-Saharan Africa with which to compare the results of this study. Nevertheless, a study on the cost of NCDs among the Kenyan general population estimated the average T1DM patient payment for a physician consultation, medication, and hypoglycaemia admission as $186.40 and $541.22 in public and private facilities, respectively. For T2DM patients, the payment for the same services decreased to $88.61 and $488.60 in public and private facilities, respectively. Patients who needed both insulin and oral medications spent $234.44 in public and $675.85 in private facilities [47]. A study analysing the direct medical cost of both T1DM and T2DM in Turkey estimated the cost at $607 [47]. Analyses including complications and co-morbidities in the same setting yielded higher unit costs [47, 48]. However, these figures are not directly comparable to our findings due to differences in costing perspectives used, settings and the inclusion of some complications.

Conversely, a study from Thailand by Riewpaiboon et al. (2007) analyzed the cost of diabetes care in a public hospital using a provider perspective and yielded a cost of $3.58 per outpatient visit, which is close to the results of this study [20]. The comparability with our study was limited as the facility used a combination of generic and branded drugs, whereas the Hagadera hospital used generics only.

The findings of this study would be relevant to humanitarian program implementers as well as to program managers budgeting for NCD services in low-income countries, as well as NCD care at the primary care level. It is also expected to also benefit donors who fund NCD care at the primary care level. Further, as this is the first study of its kind in this setting, the analysis and findings also form a baseline for future research.

## Limitations

Our study is set within a humanitarian setting where studies have shown a great risk of underdiagnosis of NCDs [49]. Hence, the findings may not be generalizable outside such contexts. Furthermore, access to insulin is very challenging within humanitarian settings and it is often

reserved for those with T1DM, indicating possible undertreatment for patients with T2DM requiring insulin. Other characteristics related to the location, such as logistical constraints caused by security situations and donor-funded operations, can impact on costs. The impact of COVID-19 on the calculated estimates was not taken into account as data collection activities took place before the pandemic. The incurred costs at the health posts were not captured in this study as it was difficult to collect data specific to diabetes follow-ups at the community level. The provider perspective adopted has excluded OOP and other societal costs. Such costs are likely significant in a country like Kenya where OOP health expenditure is estimated at more than one-fifth of the health funds of 2018 [50]. While acknowledging the likely complexity and resource requirements, a primary study adopting a societal perspective and estimating OOP would make further contributions to this evidence base. The primary drawback of using overall utilization data is the possibility of underestimating consumed resources. The estimated costs were also undervalued as complications and prevention costs were not considered [51], and the consultations did not distinguish between types of visits including lost follow-ups. A linearity between direct and indirect costs is assumed by the mark-up method of assigning overhead costs. In reality, some operations may in fact consume more or less resources than others [52]. However, the main considerations for using this method were its practicality and because a wide variation of indirect costs was not anticipated.

## Conclusions

Investment in holistic and sustainable NCD services such as prevention, early diagnosis, treatment, and follow-up strategies should be at the forefront of the humanitarian response. These services would be beneficial with immediate implications on the COVID-19 response and/or chronic care over the short-term. It can also reduce the burden, particularly, with early diagnosis to prevent complications. Cost analyses contribute critically to economic evaluations and related investment decisions. They are vital in understanding and evaluating the overall value of treatment programs, interventions, and healthcare policies [16]. Despite the listed study limitations, essential basic services for the management of uncomplicated diabetes can be offered at a modest and affordable cost. Therefore, integrating diabetes care into PHC services should be seen as a fundamental pillar of a long-term policy response by governments, agencies, international organisations, and other stakeholders. Further economic evaluations would provide the necessary evidence to bridge more gaps in information, and support advocacy efforts to raise worldwide political awareness on diabetes and other NCDs. In addition, the costing methods used in this paper can be used for other NCDs with tools adapted according to treatment pathways for each NCD. Increasing the relevance of related issues in global health, finance, development policy, political forums, and other processes would result in commitments by governments, multilateral agencies and donors to increase NCDs and Universal Health Coverage (UHC) resources for humanitarian contexts and the general public.

## Supporting information

**S1 Table. Methodology, data requirements, calculations, and sources of data.**
(DOCX)

**S2 Table. Distribution of economic costs.**
(DOCX)

## Author Contributions

**Conceptualization:** Lucy Kanya, John Kiogora, Lilian Kiapi, Caitlin Tulloch.

**Formal analysis:** Lizah Masis, Lucy Kanya, John Kiogora.

**Investigation:** Lizah Masis.

**Methodology:** Lucy Kanya, John Kiogora, Lilian Kiapi, Caitlin Tulloch.

**Project administration:** Lucy Kanya, Lilian Kiapi, Ahmad Hecham Alani.

**Resources:** John Kiogora, Caitlin Tulloch.

**Supervision:** Lucy Kanya, John Kiogora, Lilian Kiapi.

**Validation:** Lizah Masis, John Kiogora, Lilian Kiapi, Ahmad Hecham Alani.

**Visualization:** Ahmad Hecham Alani.

**Writing – original draft:** Lizah Masis, Ahmad Hecham Alani.

**Writing – review & editing:** Lizah Masis, Lucy Kanya, John Kiogora, Lilian Kiapi, Caitlin Tulloch, Ahmad Hecham Alani.

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
