## [Decision Letter · Decision Letter 0]

25 Apr 2022

PONE-D-22-03317Estimating treatment costs for uncomplicated diabetes at the International Rescue Committee Hagadera Hospital in KenyaPLOS ONE

Dear Dr. Kanya,

Thank you for submitting your manuscript to PLOS ONE. After careful consideration, we feel that it has merit but does not fully meet PLOS ONE’s publication criteria as it currently stands. Therefore, we invite you to submit a revised version of the manuscript that addresses the points raised during the review process.

Thank you very much for submitting your work to PLOS ONE. The reviewers have suggested that you revise your manuscript based on their comments to make it publishable. Please address the reviewers' concerns point-by-point and submit your revised manuscript.

Also, please address following editorial concerns as well:

1. The American Diabetes Association has suggested that the adjective 'diabetic' be not used with patients. Hence, write these as 'diabetes patients' or 'patients with diabetes'.

2. Rewrite references 27 and 35 properly.

We look forward to receiving your revised manuscript.

Kind regards,

Binaya Sapkota, PharmD

Academic Editor

PLOS ONE

Journal Requirements:

3. Please include a complete copy of PLOS’ questionnaire on inclusivity in global research in your revised manuscript. Our policy for research in this area aims to improve transparency in the reporting of research performed outside of researchers’ own country or community. The policy applies to researchers who have travelled to a different country to conduct research, research with Indigenous populations or their lands, and research on cultural artefacts. The questionnaire can also be requested at the journal’s discretion for any other submissions, even if these conditions are not met.  Please find more information on the policy and a link to download a blank copy of the questionnaire here: https://journals.plos.org/plosone/s/best-practices-in-research-reporting. Please upload a completed version of your questionnaire as Supporting Information when you resubmit your manuscript.

Additional Editor Comments (if provided):

Thank you very much for submitting your work to PLOS ONE. The reviewers have suggested that you revise your manuscript based on their comments to make it publishable. Please address the reviewers' concerns point-by-point and submit your revised manuscript.

Also, please address following editorial concerns as well:

1. The American Diabetes Association has suggested that the adjective 'diabetic' be not used with patients. Hence, write these as 'diabetes patients' or 'patients with diabetes'.

2. Rewrite references 27 and 35 properly.

Reviewers' comments:

Reviewer's Responses to Questions

**Comments to the Author**

1. Is the manuscript technically sound, and do the data support the conclusions?

Reviewer #1: Yes

Reviewer #2: Yes

2. Has the statistical analysis been performed appropriately and rigorously? 

Reviewer #1: Yes

Reviewer #2: Yes

3. Have the authors made all data underlying the findings in their manuscript fully available?

Reviewer #1: No

Reviewer #2: No

4. Is the manuscript presented in an intelligible fashion and written in standard English?

Reviewer #1: Yes

Reviewer #2: No

5. Review Comments to the Author

Reviewer #1: • General comments

This paper addresses a relevant topic, that of diabetes, and in a relevant population, that of refugees. It is well written and the methods appear rigorous enough, from what I can judge, since I am not an economist. The results are probably useful for international NGOs providing medical care to refugees. However, one can question its usefulness for other stakeholders; the authors will want to expand on this in their discussion and perhaps also in their conclusion.

• Specific comments

1. In the title, is it possible to use “hospital serving refugees in Kenya” instead of giving the name of the hospital?

2. Abstract: RBS should be defined.

3. Why refer to “holistic” management when there is no consideration of prevention or follow-up in the paper?

4. It would be useful for the readership to have an idea of the estimated prevalence of diabetes in the refugee camp.

5. In Africa, diabetes is usually diagnosed once there are already complications unless active screening takes place. Of the patients treated in 2017, how many did NOT have complications? Also, is information on the time since diagnosis available?

6. This study took place before the pandemic: this is perhaps an important point to mention.

7. If I understood correctly, it is assumed that patients with type 2 diabetes are solely on hypoglycemic drugs, not on insulin? Please explain.

8. The content of the interviews should be briefly described.

9. It would be important to know why the health posts were not included in the study and to include this as a limitation of the study.

10. Do the patients pay for consultations, tests or drugs? If everything is free for the refugees, then the findings are not applicable where patients have to pay out of their pocket, which is the case in most African countries.

11. Is it possible to explain the term “gross-costings”, and “top-down” and “bottom-up” micro-costing, for non-economists?

12. This costing method may be of value for other NCDs; the authors will want to say a word on this.

Reviewer #2: I read your paper with great interest. The topic is novel. It is better to improve the English language of the paper. There are several mistakes. Moreover, it is better to use clear language with small sentences.

Studied----study

Which---the which

As the evidence----as evidence

Which accounted----that accounted

and direct/indirect cost items --- and direct/indirect cost items are

The clinical pathway for care of uncomplicated diabetes------ The clinical pathway for the care of uncomplicated diabetes

Measurement, valuation and allocation of costs to health----- Measurement, valuation, and allocation of costs to the health

aAdapted----adapted

useful---of useful

organisational----organizational

Cost analyses contributes critically into economic evaluations.------- Cost

analyses contribute critically to economic evaluations.

Analysed----analyzed

At 2008---of 2008

İn direct costs-----indirect costs

Such---such as

World-wide----worldwide

Are the costs in your study annual or not? Please mention.

The discussion part of the paper should be improved. Please add more references. Please compare them. Please read these papers and cite them.

1. S.Ö. Keskek, S. Kırım, N. Yanmaz, N. Sahinoglu Keskek, G. Ortoglu, A. Canataroglu , "Direct medical cost of type 1 and type 2 diabetes in Turkey", International Journal of Diabetes in Developing Countries, Ekim-2013, DOI 10.1007/s13410-013-0159-6

2. Ş.Ö. Keşkek, S.Kırım, N. Yanmaz. Estimated costs of the treatment of diabetic foot ulcers in a tertiary hospital in Turkey. Pak J Med Sci 2014; 30(5): 968-971. doi: http://dx.doi.org/10.12669/pjms.305.5182

3.Gülümsek E, Keşkek ŞÖ. Direct medical cost of nephropathy in patients with type 2 diabetes. Int Urol Nephrol. 2021 Oct 18. doi: 10.1007/s11255-021-03012-

6. PLOS authors have the option to publish the peer review history of their article (what does this mean?). If published, this will include your full peer review and any attached files.

Reviewer #1: No

Reviewer #2: **Yes: **Şakir Özgür Keşkek

---

## [Author Response · Author response to Decision Letter 0]

29 Jun 2022

Responses to Editor's comments (marked R:)

1. The American Diabetes Association has suggested that the adjective 'diabetic' be not used with patients. Hence, write these as 'diabetes patients' or 'patients with diabetes'.

R: Thank you for this suggestion. Adjustments have been made throughout the manuscript. 

2. Rewrite references 27 and 35 properly.

R: Thank you for flagging this. References 27 and 35 have been amended accordingly. In addition, all references have been reviewed once more and are now in accordance with PLOS One requirements. 

Journal Requirements 

R: The revised manuscript meets PLOS ONE's style requirements. 

2. Please provide additional details regarding participant consent. 

R: Additional details are now included in the ethics statement within the Methods section.

3. Please provide a complete cop of PLOS' questionnaire on inclusivity in global research in your revised manuscript. 

R: The completed PLOS questionnaire is included with this resubmission. 

4. Regarding data.

R: All data is provided within the manuscript’s tables. We have also included supplementary tables to facilitate replicability and repeatability of the study methodology.

5. Please include your full ethics statement in the 'Methods' section of your manuscript. 

R: The full ethics statement has now been moved to the 'Methods' section. This includes the full names of the ethics review committees who approved the study and the approval numbers. As mentioned in the manuscript, informed consent was obtained from all health care providers interviewed as part of the data collection process. 

6. Reference list

R: The reference list has been reviewed and we confirm that this is complete and correct. 

Responses to Reviewers comments

Reviewer #1: • General comments

This paper addresses a relevant topic, that of diabetes, and in a relevant population, that of refugees. It is well written and the methods appear rigorous enough, from what I can judge, since I am not an economist. The results are probably useful for international NGOs providing medical care to refugees. However, one can question its usefulness for other stakeholders; the authors will want to expand on this in their discussion and perhaps also in their conclusion.

R: Thank you for your comments. We have expanded on the usefulness of these findings in the discussion section and conclusion. 

• Specific comments

1. In the title, is it possible to use “hospital serving refugees in Kenya” instead of giving the name of the hospital?

R: Thank you. Title adjustment made to “hospital serving refugees in Kenya”. 

2. Abstract: RBS should be defined.

R: Random blood sugar (RBS) tests are now defined in the abstract. 

3. Why refer to “holistic” management when there is no consideration of prevention or follow-up in the paper?

R: Our study methodology has taken into account the cost of follow-up for patients with diabetes. However, given the humanitarian context and its associated data collection challenges, we were unable to differentiate among newly diagnosed patients and those who came for a re-visit. Nevertheless, the treatment protocols followed within the facility were similar for both patients segments. However, we do acknowledge that prevention costs are not included. Holistic management in our analysis therefore refers to the care that patients receive from the point of testing, and subsequent care when the patients return to the facility for a revisit. We have included this in the revised manuscript. 

4. It would be useful for the readership to have an idea of the estimated prevalence of diabetes in the refugee camp.

R: It is estimated that the Hagadera refugee camp has nearly 1,000 patients with diabetes under follow-up in 2018. In 2017, diabetes patients represented 11.76% of the OPD consultations. However, no previous diabetes prevalence studies have been conducted in the camp. The manuscript has been updated. 

5. In Africa, diabetes is usually diagnosed once there are already complications unless active screening takes place. Of the patients treated in 2017, how many did NOT have complications? Also, is information on the time since diagnosis available?

R: This observation is correct and is likely the case in the study setting. However, diabetes complications were not captured as they were out of the scope of this paper. In addition, the study has focused on refugees; excluding the general population in Kenya. 

6. This study took place before the pandemic: this is perhaps an important point to mention.

R: Thank you. This has now been included in the paper in the limitations section.

7. If I understood correctly, it is assumed that patients with type 2 diabetes are solely on hypoglycemic drugs, not on insulin? Please explain.

R: Unfortunately, access to insulin is very challenging within humanitarian settings (and the general population in Kenya) and it is often reserved for those with type 1 diabetes. 

8. The content of the interviews should be briefly described.

R: Interviews were conducted to determine inputs required for diabetes care, and to estimate staff time allocation. The manuscript is updated.

9. It would be important to know why the health posts were not included in the study and to include this as a limitation of the study.

R: Thank you for the suggestion to include this as a limitation of our study. The incurred costs at the health posts were out of scope of the study as these services are provided by community health workers (CHWs) who also provide other services including maternal and child health, infectious diseases etc. Consequently, it was difficult to collect data specific to diabetes follow ups at the community level. We mention this in the manuscript too.

10. Do the patients pay for consultations, tests or drugs? If everything is free for the refugees, then the findings are not applicable where patients have to pay out of their pocket, which is the case in most African countries.

R: Thank you for your question. The study has followed a provider perspective approach when calculating the estimated costs. Hence, out of pocket (OOP) and other societal costs were outside the scope of the study. Such costs are likely significant in a country like Kenya where OOP health expenditure is estimated at more than one-fifth (as stated in the limitations section), however, it is important to note that there was no expenses for patients to receive services at the hospital. The analysis from a provider perspective are therefore very important for program implementers, including payers such as the government of Kenya in their plans to extend universal health care coverage to its populace. 

11. Is it possible to explain the term “gross-costings”, and “top-down” and “bottom-up” micro-costing, for non-economists?

R: Noted. This has now been included in the manuscript.

12. This costing method may be of value for other NCDs; the authors will want to say a word on this.

R: Thank you for raising this important point. We have included a statement within the conclusion demonstrating that similar costing methods would be useful for other NCDs, with tools adapted according to treatment pathways for each NCD.

Reviewer #2: I read your paper with great interest. The topic is novel. It is better to improve the English language of the paper. There are several mistakes. Moreover, it is better to use clear language with small sentences.

Studied----study

Which---the which

As the evidence----as evidence

Which accounted----that accounted

and direct/indirect cost items --- and direct/indirect cost items are

The clinical pathway for care of uncomplicated diabetes------ The clinical pathway for the care of uncomplicated diabetes

Measurement, valuation and allocation of costs to health----- Measurement, valuation, and allocation of costs to the health

aAdapted----adapted

useful---of useful

organisational----organizational

Cost analyses contributes critically into economic evaluations.------- Cost

analyses contribute critically to economic evaluations.

Analysed----analyzed

At 2008---of 2008

İn direct costs-----indirect costs

Such---such as

World-wide----worldwide

R: Thank you very much for your feedback. We have revised the manuscript accordingly and maintained consistency in the use of UK English. 

Are the costs in your study annual or not? Please mention.

R: Costs are reported as unit costs. Kindly refer to table 3 for the estimated unit costs in USD. Additional details were also added in table 1 for the annualization factors that were used where applicable. 

The discussion part of the paper should be improved. Please add more references. Please compare them. Please read these papers and cite them.

1. S.Ö. Keskek, S. Kırım, N. Yanmaz, N. Sahinoglu Keskek, G. Ortoglu, A. Canataroglu , "Direct medical cost of type 1 and type 2 diabetes in Turkey", International Journal of Diabetes in Developing Countries, Ekim-2013, DOI 10.1007/s13410-013-0159-6

2. Ş.Ö. Keşkek, S.Kırım, N. Yanmaz. Estimated costs of the treatment of diabetic foot ulcers in a tertiary hospital in Turkey. Pak J Med Sci 2014; 30(5): 968-971. doi: http://dx.doi.org/10.12669/pjms.305.5182

3.Gülümsek E, Keşkek ŞÖ. Direct medical cost of nephropathy in patients with type 2 diabetes. Int Urol Nephrol. 2021 Oct 18. doi: 10.1007/s11255-021-03012-

R: Thank you very much for sharing these references. While they are not specifically focussed on a humanitarian setting as our analysis is, we have read and included findings and conclusions from them in the discussion section of the revised manuscript.

---

## [Decision Letter · Decision Letter 1]

25 Aug 2022

PONE-D-22-03317R1Estimating treatment costs for uncomplicated diabetes at a hospital serving refugees in KenyaPLOS ONE

Dear Dr. Kanya,

Thank you for submitting your manuscript to PLOS ONE. After careful consideration, we feel that it has merit but does not fully meet PLOS ONE’s publication criteria as it currently stands. Therefore, we invite you to submit a revised version of the manuscript that addresses the points raised during the review process.

We look forward to receiving your revised manuscript.

Kind regards,

Binaya Sapkota, PharmD

Academic Editor

PLOS ONE

Journal Requirements:

Additional Editor Comments:

Thank you very much for addressing many of the concerns of the reviewers and the editor in the revision version. Still, there are few concerns raised by the reviewer to be addressed. Hence, I request the authors to address those remaining concerns and again send the revised manuscript.

Reviewers' comments:

Reviewer's Responses to Questions

**Comments to the Author**

1. If the authors have adequately addressed your comments raised in a previous round of review and you feel that this manuscript is now acceptable for publication, you may indicate that here to bypass the “Comments to the Author” section, enter your conflict of interest statement in the “Confidential to Editor” section, and submit your "Accept" recommendation.

Reviewer #1: All comments have been addressed

2. Is the manuscript technically sound, and do the data support the conclusions?

Reviewer #1: Yes

3. Has the statistical analysis been performed appropriately and rigorously? 

Reviewer #1: Yes

4. Have the authors made all data underlying the findings in their manuscript fully available?

Reviewer #1: Yes

5. Is the manuscript presented in an intelligible fashion and written in standard English?

Reviewer #1: Yes

6. Review Comments to the Author

Reviewer #1: Our comments were addressed in the rvised version. However, I still find the explanations for the types of costing unclear for non-economists. I also think that some more discussion should be devoted to out-of-pocket expenditures although this is not the approach of this study. We are surprised that there was no reference to the following paper: Alouki K, et al. Simple calculator to estimate the medical cost of diabetes in sub-Saharan Africa. World J Diabetes 2015; 6: 1312-22.

7. PLOS authors have the option to publish the peer review history of their article (what does this mean?). If published, this will include your full peer review and any attached files.

Reviewer #1: No

---

## [Author Response · Author response to Decision Letter 1]

7 Oct 2022

Dear reviewers,

We thank you very much for your time and insights towards improving our manuscript. Adjustments were made to further clarify the methodology for non-economists. Thank you for pointing us to the above paper by Alouki et al (2015). We have included this in the discussion section as suggested.

Kind regards,

Dr. Kanya

for Authors

---

## [Editor Report · Decision Letter 2]

12 Oct 2022

Estimating treatment costs for uncomplicated diabetes at a hospital serving refugees in Kenya

PONE-D-22-03317R2

Dear Dr. Kanya,

We’re pleased to inform you that your manuscript has been judged scientifically suitable for publication and will be formally accepted for publication once it meets all outstanding technical requirements.

Kind regards,

Binaya Sapkota, PharmD

Academic Editor

PLOS ONE

Additional Editor Comments (optional):

Thank you very much for submitting your valuable research to PLOS ONE. We are pleased to accept your paper based on your satisfactory response to the reviewers' comments and the editorial discretion.
---

## [Editor Report · Acceptance letter]

17 Oct 2022

PONE-D-22-03317R2 

Estimating treatment costs for uncomplicated diabetes at a hospital serving refugees in Kenya 

Dear Dr. Kanya:

I'm pleased to inform you that your manuscript has been deemed suitable for publication in PLOS ONE. Congratulations! Your manuscript is now with our production department. 

Kind regards, 

on behalf of

Dr. Binaya Sapkota 

Academic Editor

PLOS ONE